# Effectiveness of Curcumin in Reducing Self-Rated Pain-Levels in the Orofacial Region: A Systematic Review of Randomized-Controlled Trials

**DOI:** 10.3390/ijerph19116443

**Published:** 2022-05-25

**Authors:** Barbara Sterniczuk, Paul Emile Rossouw, Dimitrios Michelogiannakis, Fawad Javed

**Affiliations:** Department of Orthodontics and Dentofacial Orthopedics, Eastman Institute for Oral Health, University of Rochester, Rochester, NY 14620, USA; barbara_sterniczuk@urmc.rochester.edu (B.S.); emile_rossouw@urmc.rochester.edu (P.E.R.); dimitrios_michelogiannakis@urmc.rochester.edu (D.M.)

**Keywords:** *Curcuma longa*, dental, oral, orofacial, pain, treatment, turmeric, curcumin

## Abstract

The aim was to systematically review randomized controlled trials (RCTs) that assessed the effectiveness of curcumin in reducing self-rated pain levels in the orofacial region (OFR). The addressed focused question was “Is curcumin effective in reducing self-rated pain levels in the OFR?”. Indexed databases (PubMed (National Library of Medicine), Scopus, EMBASE, MEDLINE (OVID), and Web of Science) were searched up to and including February 2022 using different keywords. The inclusion criteria were (a) original studies (RCTs) in indexed databases; and (b) studies assessing the role of curcumin in the management of pain in the OFR. The risk of bias was assessed using the Cochrane risk of bias tool. The pattern of the present systematic review was customized to primarily summarize the pertinent information. Nineteen RCTs were included. Results from 79% of the studies reported that curcumin exhibits analgesic properties and is effective in reducing self-rated pain associated with the OFR. Three studies had a low risk of bias, while nine and seven studies had a moderate and high risk of bias, respectively. Curcumin can be used as an alternative to conventional therapies in alleviating pain in the OFR. However, due to the limitations and risk of bias in the aforementioned studies, more high-quality RCTs are needed.

## 1. Introduction

Pain in the orofacial region (OFR) is often described as pain perception of musculoskeletal, neurovascular, or neuropathic origin [1]; however, it also encompasses pain in dental and mucosal tissues caused by infection or inflammation [2,3]. Secondary etiological risk factors of orofacial pain (OFP) include nerve trauma, neurovascular disorders, and temporomandibular joint or muscular disorders [4]. In the United States, nearly 22% of individuals experience OFP in some capacity during any given 6-month period [5] with females and younger individuals between 15 and 45 years of age being most susceptible to OFP [5,6]. Multi-faceted pathophysiology and psychosocial comorbidity often challenges the correct diagnosis and management of OFP. Traditionally, OFP is treated using (a) medications such as non-steroidal anti-inflammatory drugs, local anesthetics, muscle relaxants, endocannabinoids, anti-convulsants, and antidepressants [7,8,9]; (b) oral appliances such as occlusal splints [10]; (c) massage therapy [11]; and (d) diode-laser therapy [12].

Complementary alternate medications (CAM) are usually derived from medicinal plants and are perceived to have no undesirable side effects compared with synthetic pharmacological drugs [13,14,15]. Patients often use CAM for the relief of pain including OFP [13,16].
Curcumin, a naturally occurring flavonoid
[17] chemically denoted as (1*E*,6*E*)-1,7-bis(4-hydroxy-3-methoxyphenyl)-1,6-heptadiene-3,5-dione), has two aromatic *O*-methoxy phenolic components, a β-dicarbonyl group, and a seven-carbon linker containing two enone groups [18]. It is a major constituent of several herbs including turmeric [17], which is a common Indian spice and is also consumed when health-related curcumin effects are desired [19,20]. Most turmeric extracts contain three major curcuminoids, including curcumin (60–70%), demethoxycurcumin (20–27%), and bisdemethoxycurcumin (10–15%), along with many other less abundant secondary metabolites [21,22]. Throughout the literature, there is currently a generalized lack of distinction between curcumin and turmeric, with many studies using the terms interchangeably [23,24]. It has been reported that even doses up to 12 g/day are safe [25]; however, curcumin exhibits poor bioavailability due to poor absorption, low intrinsic activity, and a high rate of metabolism and excretion [21,26]. Despite lower bioavailability, it has been reported that curcumin exhibits antioxidant, analgesic, and anti-inflammatory properties [17,27]. Therapeutic effects of curcumin have been assessed with regard to many diseases including cancer, diabetes mellitus, arthritis, neurological diseases, and Crohn’s disease [26,28,29,30,31,32]. Moreover, curcumin has been reported to exhibit anti-cariogenic and immunomodulatory characteristics [33,34]. In a randomized controlled trial (RCT), the efficacy of a curcumin-based gel was compared with a non-eugenol dressing in reducing post-operative pain following periodontal flap surgery in patients with periodontitis [35]. The study concluded that the curcumin-based gel is as effective as a non-eugenol dressing in promoting periodontal healing after flap surgery, and can therefore be used as a substitute form of periodontal post-operative dressing [35]. Similarly, results from another RCT showed that post-operative pain (assessed using the numeric rating scale [NRS]) after surgical extraction of impacted third molars (ITM) is significantly less among individuals who consume curcumin compared with patients using mefenamic acid (MA) [24]. Nevertheless, it has also been proposed that curcumin is not superior to MA in terms of reducing post-operative pain after surgical extraction of ITM [36]. This suggests that there is controversy regarding the effectiveness of curcumin in reducing pain in the OFR. Following a vigilant review of pertinent indexed literature, the authors observed that there are no studies that have systematically reviewed the effectiveness of turmeric and turmeric products in the management of pain in the OFR.

With this background, the aim of the present study was to systematically review RCTs that assessed the effectiveness of curcumin in reducing pain levels in the OFR.

## 2. Materials and Methods

### 2.1. Ethical Approval

In the present study, pertinent indexed literature was reviewed. In this context, prior approval from an institutional review board/committee was not required.

### 2.2. Focused Question, PICO, and PRISMA

The focused question “Is turmeric effective in reducing self-rated pain levels in the OFR?” was addressed using the Population, Intervention, Control, and Outcomes guidelines where P = Patients with pain in the OFR, I = management of pain with curcumin, C = pain management using sources other than curcumin or no treatment, and O = reduction in self-rated pain levels in the OFR. The Preferred Reporting Items for Systematic Reviews and Meta-analysis (PRISMA) guidelines were used during the literature search [37]. The protocol for the present systematic review was registered with PROSPERO (CRD42021278739).

### 2.3. Eligibility Criteria

The inclusion criterion was (a) RCTs in indexed databases that investigated the role of curcumin in the management of pain in the OFR. Commentaries, case reports, case series, letters to the Editor, and review articles (narrative and systematic) were excluded.

### 2.4. Data Sources and Search Strategy

The indexed databases (PubMed (National Library of Medicine), Scopus, EMBASE, MEDLINE (OVID), and Web of Science) were electronically searched without language and/or time barriers up to and including February 2022. A customized search strategy was developed by one author (BS): “[(pain) AND (curcumin OR turmeric) AND (orofacial OR face OR facial OR dental OR oral mucosa OR tooth OR teeth OR maxilla OR mandible OR temporomandibular joint)]”. The titles and abstracts of identified studies using the aforementioned search strategy were independently screened by two authors (BS, ER). The full texts of pertinent studies were independently reviewed, and reference lists of the relevant studies were hand-searched for any additional studies. The guidelines of the preferred reporting outcomes for systematic reviews and meta-analysis (PRISMA) were used during the literature search [38]. Disagreements in the study selection process were resolved via discussion.

### 2.5. Data Extraction

Data extraction from the nineteen eligible RCTs [24,35,39,40,41,42,43,44,45,46,47,48,49,50,51,52,53,54,55] was performed by two authors (BS, ER), and pertinent information was summarized according to the (a) reference, (b) number of subjects, (c) subject gender, (d) study groups, (e) subjects’ mean age and range, (f) source of OFP, (g) pain rating scale, (h) follow-up period, (i) study characteristics related to curcumin use, (j) type of control, (k) risk of bias, and (l) main study outcomes (reduction in self-rated pain levels) and conclusions.

### 2.6. Risk of Bias Assessment

The risk of bias (RoB) in the included studies was assessed using the Cochrane RoB tool [56]. The following parameters were used to assess the RoB: (a) Random sequence generation; (b) allocation concealment; (c) selective reporting (based on the availability of pre-specified primary and secondary outcomes); (d) blinding of investigators and participants; (e) blinding of outcome assessment; (f) incomplete outcome data; and (g) other bias due to problems not covered in the study. The RoB in each category was assessed as “low-risk”, “high-risk”, or “unclear-risk”, with the last category indicating either lack of information or uncertainty over the potential for bias [56]. Based on the criteria, each study was determined to have either a low, moderate, or high overall RoB.

## 3. Results

The initial search of electronic databases showed 593 studies and 15 studies from the hand search. After duplicates were removed, 524 studies remained. Five hundred and five studies that did not fulfill the eligibility criteria were excluded. In total, nineteen studies [24,35,39,40,41,42,43,44,45,46,47,48,49,50,51,52,53,54,55] were included and processed for data extraction (Figure 1).

### 3.1. General Characteristics of the Studies Included

Nineteen RCTs were included [24,35,39,40,41,42,43,44,45,46,47,48,49,50,51,52,53,54,55]. The number of participants ranged between 11 and 178 individuals. Fifteen studies [24,35,39,41,42,43,44,45,46,48,49,50,51,52,54] reported the gender of the participants in which the number of males and females ranged from 15 to 44 and 9 to 46 individuals, respectively. In all studies [24,35,39,40,41,42,43,44,45,46,47,48,49,50,51,52,53,54,55], individuals in the test group used turmeric products for the management of pain in the OFR. In twelve studies [39,41,43,45,46,48,50,52,53,54], individuals in the control group used steroids for the management of pain in the OFR. In three studies, study groups were compared to a placebo [42,44,51]. In the studies by Maulina et al. [24] and Lone et al. [40], participants used MA and zinc-oxide eugenol, respectively, for the management of pain in the control group. Meghana et al. [35] compared the placement of curcumin gel in the study group to the placement of COE-pak in controls. The mean age of participants was reported in thirteen studies [35,39,41,42,43,44,45,46,48,50,52,54]; mean ages ranged from 9.62 ± 43.72 to 65.2 ± 9.3 years. In fourteen studies [35,39,41,42,43,45,46,47,48,49,50,53,54,55], pain was assessed using the visual analogue scale (VAS), a continuous scale 100 mm in length, anchored on each end by a descriptor for each pain extreme such as “no pain” and “worst imaginable pain”; respondents receive a score up to a 100 for the linear distance they indicate along the scale [57]. In four studies [24,44,51,52], pain was assessed using the numeric rating scale, a segmented numeric version of the VAS in which participants select a whole number from zero to ten that best represents the intensity of pain [57].

In studies by Maulina et al. [24] and Lone et al. [40], pain in the OFR was associated with the extraction of impacted mandibular third molars and alveolar osteitis (AO), respectively. One study evaluated pain from healing extraction sockets in type II diabetics [55]. Two studies examined pain from periodontal flap surgery [35,51]. Four studies [39,45,52,53] evaluated the effect of turmeric on pain from recurrent aphthous stomatitis (RAS) and six studies [43,46,47,48,49,50] reported its effect on pain from oral lichen planus (OLP).

In the study by Mansourian et al. [41], pain in the OFR was associated with graft vs. host disease. In the study by Nakao and colleagues, oral pain in patients after head and neck radiotherapy was assessed [42], while chemotherapy-induced oral mucositis with and without head and neck radiotherapy was assessed by Kia et al. [44]. These results are shown in Table 1.

### 3.2. Curcumin-Related Parameters

Curcumin administered to study groups included systemic curcumin in oral capsule or tablet form in four studies [24,43,44,50], and a lozenge form in one study [54]. Topical curcumin products were used in the form of oral gel or paste in ten studies [35,39,41,42,45,46,47,48,49,52], topical dressing in three studies [40,51,55], and topical powder in one study [53]. Eight studies administered curcumin products in combination with another substance to their study group participants, including amoxicillin [24], amoxicillin with diclofenac [51], prednisone and cyclosporine [41], mustard oil [40], dexamethasone with nystatin [50], ibuprofen [35], hifenac with novamox [55], and clove oil [54]. The study duration ranged between 24 h [24] and 3 months [54] (see Table 1). The frequency of curcumin therapy ranged from the continuous application (as a topical dressing) [40,48,55] to once daily [42] (Table 2 and Table 3). One study did not report the frequency of application [41]. The concentration of curcumin given in capsule or tablet form ranged between 80 mg per day [43] to 2000 mg per day [50]. Topical curcumin concentration in gel or paste form ranged from 160 μg/mL in one study [42] to 5% curcumin in others [45,46] and included concentrations of 10 mg/gram [39], 0.5% curcumin [51], and 2% curcumin [52]. Eight studies did not report the concentration of curcumin delivered [35,40,41,47,48,49,53,55] (Table 2 and Table 3).

#### 3.2.1. Outcomes of Included Studies

In ten of the studies, pain scores of test and control subjects were directly compared to baseline [24,39,42,43,45,48,50,52,53,54]; in nine studies, it was found that pain scores were significantly reduced compared to baseline in both curcumin groups and control groups that were administered an active substance [24,39,43,45,48,50,52,53,54], while in one study comparing curcumin to a placebo, pain scores were not significantly different from baseline for either test or control groups in patients undergoing head and neck radiotherapy [42]. In contrast, Kia et al. [44] found that curcumin nanomicelle capsules were more effective compared to a placebo in reducing oral pain scores in patients undergoing chemotherapy either with or without head and neck radiotherapy. Anil et al. [51] also found pain scores to be significantly decreased in subjects receiving a curcumin mucoadhesive film vs. a placebo film after undergoing periodontal flap surgery. Eleven studies compared curcumin therapy with corticosteroid therapy for pain reduction [39,41,43,45,46,47,48,49,50,52,53]; seven of these found no significant difference between pain scores with curcumin therapy as compared to corticosteroid therapy, showing similar efficacy in pain reduction with either therapy [39,41,43,45,46,48,53]. Two of these studies compared combined corticosteroid and curcumin therapy vs. curcumin therapy alone and found that topical curcumin with corticosteroid had a significantly greater effect on reducing pain scores than curcumin alone [49] or corticosteroid therapy alone [47]. One study found that corticosteroid therapy reduced self-rated pain scores more rapidly as compared to curcumin therapy [52], while another study found that combined curcumin and corticosteroid therapy showed no difference in reduction of pain scores compared to corticosteroid therapy alone [50]. Srivastava and colleagues [54] measured pain scores with the use of curcumin and clove oil lozenges in test subjects compared to intralesional infiltration of dexamethasone and hyaluronidase in controls and found both groups revealed the absence of pain associated with the lesion after 3 months; however, no difference between the results of the two groups was noted.

Two studies found a statistically significant result when comparing subjects’ need for NSAID analgesics as a rescue drug in the curcumin group as compared to control groups after periodontal flap surgery, with curcumin test groups requiring fewer analgesic tablets compared with groups receiving placebo with COE-pak [35,51]. In addition, the study by Maulina and colleagues [24] discovered that patients in the curcumin test group experienced significantly lower pain scores compared to the controls using mefenamic acid and concluded that systemic curcumin was an effective agent for the management of inflammatory pain after the extraction of third molars [24]. In the study by Lone et al., a topical turmeric dressing showed a greater efficacy in resolving alveolar osteitis symptoms, with test subjects experiencing symptoms for a significantly different number of days than control subjects using a Zinc oxide eugenol (ZOE) dressing [40]. It was also found by Mugilan and colleagues [55] that pain scores were significantly lower following tooth extraction on the seventh day in the group receiving the curcumin dressing compared to no dressing in diabetic patients (Table 4 and Table 5).

#### 3.2.2. Risk of Bias Assessment

In the present study, three studies had a low risk of bias [43,44,50]. Nine studies had a moderate risk of bias [35,39,41,42,45,46,51,54,55], and seven studies were determined to have a high risk of bias [24,40,47,48,49,52,53] (Table 6).

## 4. Discussion

In the present study, only RCTs were considered eligible for inclusion as they are the highest level of scientific evidence for interventional clinical studies [58]. In summary, a vigilant review of pertinent indexed literature showed 19 RCTs [24,35,39,40,41,42,43,44,45,46,47,48,49,50,51,52,53,54,55] that addressed the question in focus. Results from 79% of the studies [24,35,39,40,41,43,44,45,46,47,48,51,53,54,55] showed that curcumin exhibits analgesic properties and is effective in reducing self-rated pain in the OFR. This suggests that curcumin is a potent alternative herbal substitute for traditional pharmacological medications (such as MA) for the management of pain in the OFR. However, such a statement should be cautiously interpreted as several factors may have influenced the outcomes. Most importantly, an inconsistency was observed in the methodology of the RCTs [24,35,39,40,41,42,43,44,45,46,47,48,49,50,51,52,53,54,55] assessed. For instance, Maulina et al. [24] compared the analgesic effectiveness of 200 mg curcumin with 500 mg of MA following third molar extraction, whereas Mugilan and colleagues [55] compared the analgesic effectiveness of curcumin dressing with controls (no dressing) in the healing of extraction sockets in type II diabetic patients. In both studies [24,55], all participants received postoperative antibiotic cover as well. It has been reported that the use of antibiotics after the extraction of third molars helps reduce post-operative pain and swelling at the surgical sites [59]. Similarly, results by Halpern and Dodson [60] showed that postoperative penicillin therapy helps reduce pain and inflammation after the extraction of third molars. Therefore, it is likely that postoperative use of amoxicillin by all patients in the studies by Maulina et al. [24] and Mugilan et al. [55] contributed to reducing postoperative pain in all patients. Hence, from the authors’ perspective, it is premature to solely credit curcumin with pain relief in these studies [24,55]. Such a discrepancy could have been addressed via the inclusion of another group of patients in whom no post-operative antibiotics are prescribed. However, from a health, as well as ethical, perspective, such a study design could be demanding to implement. According to Lautenbacher et al. [61], advancing age reduces pain sensitivity in the head region. A review of the included RCTs [24,35,39,40,41,42,43,44,45,46,47,48,49,50,51,52,53,54,55] showed inconsistency in the mean ages of participants. Collectively, of those RCTs that reported an age range, the age of subjects ranged between 13 and 83 years [24,39,41,42,46,48,49,52,54]. Due to the wide range in age of the subjects assessed in most of the studies, it is necessary to postulate that the reported pre- and post-treatment pain scores precisely represent the patient population. Moreover, the source of pain varied among the studies [24,35,39,40,41,42,43,44,45,46,47,48,49,50,51,52,53,54,55]. Six studies [43,46,47,48,49,50] examined patients that had pain associated with OLP, whereas Maulina et al. [24] assessed patients with pain after exodontia. Furthermore, there was an inconsistency in the mode of curcumin delivery. In studies by Naik et al. [47,49], curcumin was delivered topically, whereas in another study, curcumin was orally administered to patients [24]. Furthermore, the concentration, dosage, and mode of curcumin delivery also varied markedly in the RCTs assessed [24,35,39,40,41,42,43,44,45,46,47,48,49,50,51,52,53,54,55]. For instance, in ten studies [35,39,41,42,45,46,47,48,49,52], curcumin was administered in gel or paste form; however, five studies [35,41,47,48,49] did not report the concentration or dosage of curcumin delivered via gel/paste form. The composition of the control gel/paste was also inconsistent among the studies [35,39,41,42,45,46,47,48,49,52]. Currently, there are no standardized guidelines for curcumin usage in terms of the mode of delivery, concentration, and frequency of use.

The RoB has been described as a deviation or systematic error in the reported results [62]. In other words, the RoB is useful in estimating the extent to which the study design and methodology minimize potential biases [62]. In general, many RCTs had moderate and low RoB [35,39,41,42,45,46,51,54,55]. However, some studies presented with high RoB [24,40,47,48,49,52,53]. For instance, Maulina et al. [24] reported that the operator who performed the extraction of impacted third molars was blinded to the medications prescribed to patients, and both medications (curcumin and MA) looked alike; however, the authors also stated that patients in the test groups received two capsules of curcumin to manage post-operative pain compared with the control group, in which the patients orally ingested one MA tablet. Another factor that possibly biased the outcomes of the RCTs [24,35,39,40,41,42,43,44,45,46,47,48,49,50,51,52,53,54,55] assessed is the wide variation in the follow-up duration, which ranged from 1 day to nearly 3 months after treatment.

The authors intended to perform a quantitative assessment (meta-analysis) of the included RCTs; however, following meticulous scrutiny of the methodology of the included studies, it is necessary to quantitatively assess the reported results and justify the aforementioned proposal. The authors carefully reviewed the methodology of the included RCTs [24,35,39,40,41,42,43,44,45,46,47,48,49,50,51,52,53,54,55] in order to determine the self-rated pain levels/scores reported by patients prior to the initiation of curcumin or alternate medication therapy. A disharmony was observed in relation to the scales used for self-rated pain assessment as 63% studies used the VAS [39,41,42,43,45,46,48,49,50,53,54,55], whereas 5% used a modified VAS [35,47] and 21% used the NRS [24,44,51,52]. Lone et. al. [40] did not report the scale used to assess self-rated pain. Therefore, standardization of baseline pain levels could not be performed, which otherwise may have assisted in comprehending pain intensities after pharmacologic and curcumin-based medications.

## 5. Conclusions

Preliminary data suggest that curcumin can be used as an alternative to conventional therapies in alleviating pain in the OFR. However, due to the limitations and risk of bias in the aforementioned studies, high-quality RCTs with a lower risk of bias are needed to further investigate curcumin use in pain management in the OFR prior to consideration for widespread clinical use.

## Figures and Tables

**Figure 1 ijerph-19-06443-f001:**
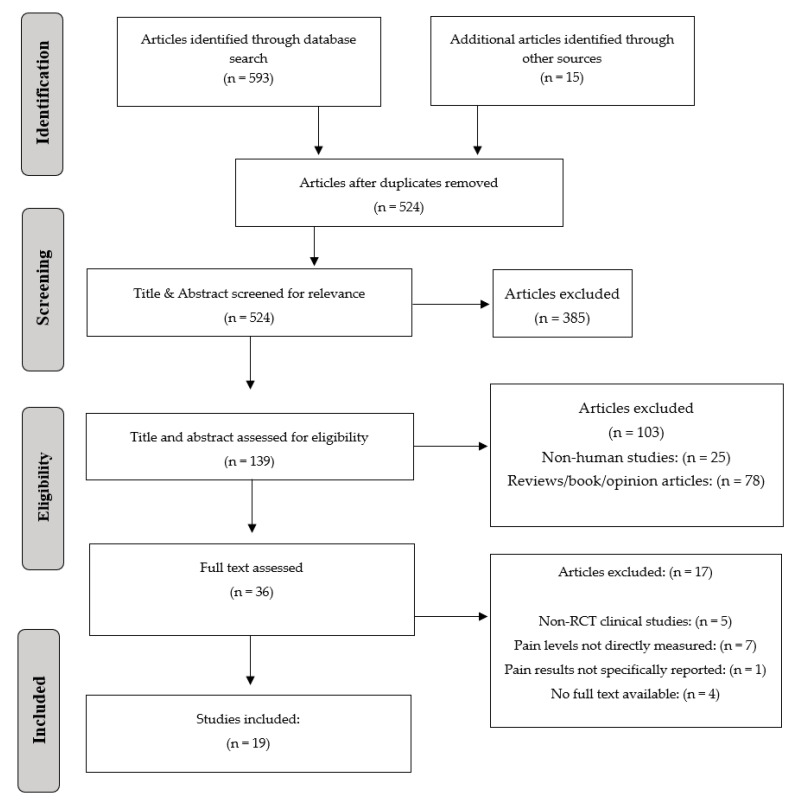
Prisma flow chart.

**Table 1 ijerph-19-06443-t001:** General characteristics of the included randomized controlled trials.

Authors et al.	Subjects (n)	Gender	Study Groups	Mean Age	Source of OFP	Scale for Rating Pain	Follow-Up
Maulina et al. [24]	90 subjects	44 males 46 females	Test group: Individuals using curcumin (45) Control-group: Individuals using MA (45)	Mean age: NR Range: 18 to 40 years	Extraction of impacted third molars	NRS	After 24 h
Meghana et al. [35]	20 subjects (40 quadrants)	Male = 7 Female = 13	Test group: Individuals receiving curcumin gel (*n* = 20) Control group: Individuals receiving periodontal dressing (*n* = 20)	All subjects mean age: 38.3 ± 9.82	Periodontal flap surgery	Modified VAS	1 week
Deshmukh et al. [39]	60 subjects	31 males 29 females	Test group: Individuals using curcumin (*n* = 30) Control group: Individuals using steroids (*n* = 30)	Mean age: 32.51 ± 11.797 years Range: 13 to 66 years	RAS	VAS	7 days
Lone et. al. [40]	178 subjects	NR	Test group: Individuals using curcumin in mustard oil (*n* = 90) Control group: Individuals using ZOE (*n* = 88)	NR	Alveolar osteitis	NR	NR
Mansourian et al. [41]	26 subjects	15 males 11 females	Test group: Individuals using curcumin (*n* = 13) Control group: Individuals using steroids (*n* = 13)	Mean age test group: 35.23 ± 7.67 years Mean age control group: 39.15 ± 12.13 years Range: 20 to 58 years	Graft vs. Host Disease *	VAS	28 days
Nakao et al. [42]	25 subjects **	16 males 9 females	Test group: Individuals using turmeric (*n* = 4) Control group: Individuals using placebo (*n* = 3)	Mean age test group: 53.4 ± 15.5 years Test group range: 28–55 years Mean age control group: 65.2 ± 9.3 years Control group range: 55–77 years	Head and neck radiotherapy	VAS	1 month (mean intervention period = 37.5 ± 11.5 days)
Kia et al. [43]	57 subjects	48 males 9 females	Test group: Individuals using nano-curcumin (*n* = 29) Control group: Individuals using prednisolone (*n* = 28)	Mean age test group: 51.86 ± 9.94 Mean age control group: 53.67 ± 8.90 Range: NR	OLP	VAS	Four weeks
Kia et al. [44]	50 subjects	28 males 22 females	Test group: Individuals receiving curcumin (*n* = 25) Control group: Individuals receiving placebo (*n* = 25)	Mean age test group: 54.98 Mean age control group: 56.94 Mean age all subjects: 55.96 ± 1.10	Chemotherapy-induced oral mucositis with and without head and neck radiotherapy	NRS	7 weeks
Kia et al. [45]	58 subjects	36 males 22 females	Test group: Individuals receiving curcumin (*n* = 29) Control group: Individuals receiving triamcinolone (*n* = 29)	Mean age test group: 9.62 ± 43.72 Mean age control group: 45.05 ± 8.9	RAS	VAS	10 days
Kia et al. [46]	50 subjects	14 males 36 females	Test group: Individuals using curcumin (*n* = 25) Control group: Individuals using triamcinolone (*n* = 25)	Mean age test group: 49.24 ± 8.17 Mean age control group: 52.08 ± 9.20 All subjects’ range: 38–73	OLP	VAS	Four weeks
Naik et al. [47]	68 subjects	NR	Test group: Individuals using curcumin gel (*n* = 34) Control group: Individuals using curcumin gel and prednisone (*n* = 34)	NR	OLP	Modified VAS	20 days
Nosratzehi et al. [48]	40 subjects	26 females 14 males; subjects matched for age and gender and divided into 2 groups	Test group: Individuals using curcumin (*n* = 20) Control group: Individuals using betamethasone with nystatin suspension (*n* = 20)	Mean age test group: 41.9 ± 11.22 Mean age control group: 38.5 ± 7.03 All subjects: 28–60 years	OLP	VAS	12 weeks
Naik et al. [49]	60 subjects	Males: 30 Females: 30	Test group: Individuals using curcumin gel only (*n* = 30) Control group: Individuals using curcumin gel and prednisolong (*n* = 30)	Range test group: 13–61 Range control group: 21–65 Total range: 13–65 Mean age: NR	OLP	VAS	20 days
Amirchaghmaghi et al. [50]	20 subjects	7 male 13 female	Test group: Individuals using curcumin tablets and dexamethasone/nystatin mouthwash (*n* = 12) Control group: Individuals using placebo and dexamethasome/nystatin mouthwash (*n* = 8)	Mean test group: 49.42 ± 11.22 Mean control group: 52.75 ± 9.43	OLP	VAS	Four weeks
Anil et al. [51]	15 subjects (30 sites)	7 males 8 females	Test group: Individuals receiving curcumin (*n* = 15) Control group: Individuals receiving placebo (*n* = 15)	All subjects mean age: 42.27 ± 6.55	Periodontal flap surgery	NRS	48 h ***
Raman et al. [52]	60 subjects	19 males 41 females	Test group: Individuals receiving curcumin (*n* = 30) Control group: Individuals receiving triamcinolone (*n* = 30)	All subjects range: 18–30 Mean age test group: 21.3 Mean age control group: 21.6	RAS	NRS	8 days ****
Halim et al. [53]	20 subjects	NR	Test group: Individuals receiving turmeric (*n* = 10) Control group: Individuals receiving triamcinolone (*n* = 10)	NR	RAS	VAS	5 days
Srivastava et al. [54]	80 subjects	71 males 9 females	Test group: Individuals receiving curcumin with clove oil (*n* = 40) Control group: Individuals receiving dexamethasone with hyaluronidase (*n* = 40)	Mean age all subjects: 33.5 ± 9.5. Range all subjects: 31–40.	Oral submucous fibrosis	VAS	3 months
Mugilan et al. [55]	11 subjects	NR	Test group: Individuals receiving curcumin dressing (*n* = 6)	NR	Extraction socket healing in Type II diabetics	VAS	7 days

MA: Mefenamic acid. NR: Not reported. NRS: Numeric rating scale. OLP: Oral lichen planus. RAS: Recurrent aphthous stomatitis. VAS: Visual analogue scale. OFP: Orofacial pain. ZOE: Zinc oxide eugenol. * Includes patients diagnosed with oral graft vs. host disease after treatment of acute myeloid leukemia, acute lymphoblastic leukemia, multiple myeloma, and Hodgkins lymphoma. ** Total number of subjects includes subjects in all groups, including those unrelated to the focused question. *** Pain outcomes were measured over 48 h. Other outcomes unrelated to focused question were measured over 7 days. **** Pain outcomes were measured over 8 days. Other outcomes unrelated to focused question were measured over 6 months.

**Table 2 ijerph-19-06443-t002:** Curcumin-related characteristics in studies with curcumin-only study groups.

Authors et al.	Condition	Curcumin (Mode of Use)	Control (Mode of Use)	Curcumin (Concentration)	Control (Concentration)	Curcumin (Frequency of Use)	Control (Frequency of Use)
Deshmukh et al. [39]	RAS	Oral gel	Oral gel	10 mg curcumin/gram	0.1% triamcinolone	Three × daily for 7 days	Three × daily for 7 days
Nakao et al. [42]	Head and neck radiotherapy	Oral gel	Oral gel	160 μg/mL in oral moisturizing gel	Placebo	Once daily for 1 month	Once daily for 1 month
Kia et al. [43]	OLP	Capsule	Capsule	80 mg nano-curcumin	10 mg prednisolone	1 cap daily for 4 weeks	1 cap daily for 4 weeks
Kia et al. [44]	Chemotherapy-induced oral mucositis	Capsule	Capsule	80 mg nanomicelle curcumin	Placebo	2 × daily	2 × daily
Kia et al. [45]	RAS	Oral gel	Oral gel	5% curcumin	0.1% triamcinolone	3 × daily	3 × daily
Kia et al. [46]	OLP	Oral paste	Oral paste	5% curcumin	0.1% triamcinolone	3 × daily for four weeks	3 × daily for four weeks
Naik et al. [47]	OLP	Oral gel	Paste of crushed tablet	Curcumin-concentration NR	Curcumin with Prednisone– concentration NR	3 × daily for 15 min	3 × daily for 15 min
Nosratzehi et al. [48]	VAS	Mucoadhesive paste	Lotion/Suspension	Curcumin-concentration NR	0.1% Betamethasone and nystatin suspension (concentration NR)	3 × daily	3 × daily
Naik et al. [49]	OLP	Oral gel	Paste of crushed tablet/oral gel mix	Curcumin–concentration NR	10 mg/tab prednisolone with	3 × daily for 15 min	3 × daily for 15 min
Raman et al. [52]	RAS	Oral gel	Oral paste	2% *Curcuma longa*-10 mg	0.1% triamcinolone	3 × daily	3 × daily
Halim et al. [53]	RAS	Powder	NR	NR	0.1% triamcinolone	2 × daily for 5 min	2 × daily for 5 min

NR: Not reported. N/A: Not applicable. RAS: Recurrent aphthous stomatitis. OLP: Oral lichen planus.

**Table 3 ijerph-19-06443-t003:** Characteristics of studies related to curcumin-combination treatments of study groups.

Authors et al.	Condition	Curcumin (Mode of Use)	Control (Mode of Use)	Curcumin (Concentration)	Control (Concentration)	Curcumin (Frequency of Use)	Control (Frequency of Use)
Maulina et al. [24]	Extraction of impacted third molars	Capsule	Capsule	500 mg amoxicillin and 200 mg curcumin	500 mg amoxicillin and 500 g mefenamic acid	2 caps every 8 h for 24 h	1 cap every 8 h for 24 h
Meghana et al. [35]	Periodontal flap surgery	Oral gels and ibuprofen tablet	Periodontal dressing and ibuprofen tablet	Curcumin-concentration NR 600 mg ibuprofen	COE-pak-concentration N/A 600 mg ibuprofen	Curcumin: Twice daily for 1 week Ibuprofen: 1 tablet every 8 h for 24 h and as needed thereafter	COE-pak: N/A Ibuprofen: 1 tablet every 8 h for 24 h an D as needed thereafter
Lone et al. [40]	Alveolar osteitis	Topical dressing	Topical dressing	Fresh ground turmeric in mustard oil-concentration NR	ZOE-concentration NR	Changed on alternate days until symptoms subsided	Changed on alternate days until symptoms subsided
Mansourian et al. [41]	Graft vs. Host Disease *	Oral gel	Oral gel	Curcumin in orabase-concentration NR Systemic prednisone and cyclosporine-concentration NR	Triamcinolone-concentration NR Systemic prednisone and cyclosporine-concentration NR	28 days; frequency NR	28 days; frequency NR
Amirchaghmaghi et al. [50]	OLP	Tablet and mouthwash	Tablet and mouthwash	1000 mg curcumin and 0.5 mg dexamethasone with nystatin suspension 100,000 units	Placebo and 0.5 mg dexamethasone with nystatin suspension 100,000 units	Two 500 mg tablets, twice daily Mouthwash three times daily	Four tablets, twice daily Mouthwash three times daily
Anil et al. [51]	Periodontal flap surgery	Curcumin mucoadhesive film	Placebo mucoadhesive film	0.5% curcumin Amoxicillin 500 mg Diclofenac 100 mg	Placebo Amoxicillin 500 mg Diclofenac 100 mg	Placed under COE-pak for 7 days	Placed under COE-pak for 7 days
Srivastava et al. [54]	Oral submucous fibrosis	Lozenge	Intralesional infiltration	100 mg curcumin, 10 mg clove oil	8 mg dexamethasone, 1500IU hyaluronidase, 0.5 mL 2% lignocaine	3 times daily	2 times per week
Mugilan et al. [55]	Extraction socket healing in Type II diabetics	Dressing	None	NR (Abbott Curenext gel) Hifenac (analgesic) concentration NR and novamox 500 mg	Hifenac (analgesic) concentration NR and novamox 500 mg	Placement immediately after extraction	N/A

ZOE: Zinc-oxide eugenol. OLP: Oral lichen planus. N/A: Not applicable. NR: Not reported. ***** All subjects with graft vs. host disease were under treatment with concomitant prednisolone and cyclosporine as part of an anti-rejection regimen.

**Table 4 ijerph-19-06443-t004:** Main outcomes and conclusions of studies with curcumin-only-treated study groups.

Authors et al.	Main Outcomes	Conclusions
Deshmukh et al. [39]	Test group ↓ vs. baseline * Control group ↓ vs. baseline Test group = control group at any time point measured	Curcumin gel showed a similar efficacy to triamcinolone gel in the treatment of minor RAS.
Nakao et al. [42]	Test group = control (placebo) groups = baseline	Turmeric in oral gel does not effectively relieve oral pain after head and neck radiotherapy.
Kia et. al. [43]	Test group ↓ vs. baseline Control group ↓ vs. baseline Test group = control group at any time point measured	Systemic curcumin showed a similar efficacy to systemic prednisone in the treatment of OLP.
Kia et. al. [46]	Test group = control group at any time point measured	Topical curcumin showed a similar outcome to topical triamcinolone in the treatment of OLP.
Naik et al. [47]	Control group ↓ vs. test group	Topical curcumin with prednisone is more effective than topical curcumin alone in the treatment of OLP.
Nosratzehi et al. [48]	Test group ↓ vs. baseline Control group ↓ vs. baseline Test group = control group at any time point measured	Topical curcumin showed a similar outcome to topical betamethasone with nystatin suspension in the treatment of OLP.
Naik et al. [49]	Test group ↓ vs. baseline Control group ↓ vs. baseline Control group ↓ vs. test group	Topical curcumin with prednisolone is significantly more effective in reducing pain compared to topical curcumin alone in the treatment of OLP.
Raman et al. [52]	Test group ↓ vs. baseline Control group ↓ vs. baseline Significantly more subjects had alleviation of pain symptoms in the control group on the first, second, third, fourth and fifth days compared to the test group	Triamcinolone paste reduces self-rated pain scores from recurrent aphthous ulcers more rapidly as compared to curcumin gel.
Kia et al. [45]	Test group ↓ vs. baseline Control group ↓ vs. baseline Test group = control group on the first, fourth, seventh or tenth days	5% Curcumin in orabase is as effective as 0.1% triamcinolone in reducing pain from aphthous ulcers.
Halim et al. [53]	Test group ↓ vs. baseline Control group ↓ vs. baseline Test group = control group on the first and fifth days of treatment	Turmeric powder and 0.1% triamcinolone had similar efficacy in reducing pain from aphthous ulcers.
Kia et al. [44]	Test group ↓ vs. control (placebo) group at week 7 in patients with and without head and neck radiotherapy Test group ↓ vs. control (placebo) group in second, fourth and seventh weeks compared to placebo in patients receiving chemotherapy only	Curcumin capsules were effective in decreasing pain in patients undergoing chemotherapy either with or without head and neck radiotherapy.

=: Pain scores not significantly different; ↓: Pain scores significantly reduced; OLP: Oral lichen planus; GVHD: Graft vs. host disease; RAS: OLP: Oral lichen planus; GVHD: Graft vs. host disease; RAS: Recurrent aphthous ulcers. * The term baseline is used to describe pain levels at the beginning of the trials.

**Table 5 ijerph-19-06443-t005:** Main outcomes and conclusions of studies with curcumin-combination-treated study groups.

Authors et al.	Main Outcome	Conclusions
Maulina et al. [24]	Test group ↓ vs. baseline * Control group ↓ vs. baseline Test group ↓ vs. baseline	Curcumin with amoxicillin is more effective for pain management after exodontia than mefenamic acid with amoxicillin.
Lone et. al. [40]	Number of days that subjects experienced symptoms of alveolar osteitis was significantly less in test vs. control groups	Curcumin dressing with mustard oil showed greater efficacy at subsiding symptoms of alveolar osteitis compared to ZOE dressing.
Amirchaghmaghi et al. [50]	Test group ↓ vs. baseline Control group ↓ vs. baseline Test group = control group	Systemic curcumin had no detectable effect in the treatment of OLP in the presence of corticosteroid therapy with dexamethasone and nystatin suspension mouth rinse.
Meghana et al. [35]	Test group = control group	Both periodontal dressing and curcumin have a positive effect on pain control after periodontal flap surgery.
Srivastava et al. [54]	Test group ↓ vs. baseline Control group ↓ vs. baseline Test group = control group after 3 months of treatment	Curcumin with clove oil is an effective alternative treatment with similar pain outcomes when compared to intralesional dexamethasone and hyaluronidase infiltration in the treatment of oral submucous fibrosis.
Anil et al. [51]	Test group ↓ vs. control group for several time points measured The number of analgesics required by the test group was significantly less than that needed by the control (placebo) group	Curcumin mucoadhesive film showed greater analgesic properties in the presence of an amoxicillin regimen as compared to placebo for periodontal post-surgical pain control.
Mugilan et al. [55]	Test group ↓ vs. control group on the 7th day	Curcumin oral gel dressing post-extraction in diabetic patients showed slightly greater potential for pain reduction in the presence of a hifenac and novamox regimen.
Mansourian et al. [41]	Pain scores were not significantly different between test and control groups at any time point measured.	Curcumin gel showed a similar efficacy to triamcinolone gel in the presence of systemic prednisone and cyclosporine for the treatment of GVHD.

=: Pain scores not significantly different; ↓: Pain scores significantly reduced; ZOE: Zinc-oxide eugenol. OLP: Oral lichen planus. * The term “baseline” is used to describe pain levels at the beginning of the trials.

**Table 6 ijerph-19-06443-t006:** Risk of bias assessment using the Cochrane risk of bias tool.

Domain	Maulina et al. [24]	Meghana et al. [35]	Deshmukh et al. [39]	Lone et al. [40]	Mansourian et al. [41]	Nakao et al. [42]	Kia et al. [43]	Kia et al. [46]	Naik et al. [47]	Nosratzehi et al. [48]	Naik et al. [49]	Amirchaghmaghi et al. [50]	Anil et al. [51]	Raman et al. [52]	Kia et al. [45]	Halim et al. [53]	Kia et al. [44]	Srivastava et al. [54]	Mugilan et al. [55]
1	○	○	○	‡	○	○	○	○	‡	●	‡	○	○	○	○	‡	○	○	○
2	○	●	○	‡	○	○	○	‡	‡	●	‡	○	○	‡	‡	‡	○	●	‡
3	●	●	‡	●	○	○	○	○	‡	●	●	○	●	●	‡	●	○	●	●
4	‡	●	‡	‡	‡	‡	○	○	‡	●	‡	‡	●	●	●	●	○	●	●
5	●	○	○	‡	○	●	‡	○	○	○	‡	○	○	●	●	○	○	○	○
6	●	○	‡	●	‡	○	○	○	●	○	○	○	●	●	○	λ	○	○	○
7	●	●	●	○	●	●	○	●	●	●	●	○	●	●	○	●	○	●	○
Summary	●	◗	◗	●	◗	◗	○	◗	●	●	●	○	◗	●	◗	●	○	◗	◗

Domains examined: 1: Random sequence generation 2: Allocation concealment, 3: Blinding of participants and researchers, 4: Blinding of outcome assessment, 5: Incomplete outcome data, 6: Selective outcome reporting, 7: Other bias. ●: High; ◗: Moderate; ○: Low; ‡: Unclear.

## Data Availability

Data are available on reasonable request.

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
