# Peer review of "Effectiveness of Curcumin in Reducing Self-Rated Pain-Levels in the Orofacial Region: A Systematic Review of Randomized-Controlled Trials"

_ijerph, 2022, doi:10.3390/ijerph19116443_

Round 1

Reviewer 1 Report

This research aims to determine if curcumin can decrease self-rated pain levels in the orofacial region by conducting a systematic review of randomized controlled trials.

This article is well written but a few comments are suggested as follows;

-More detailed explanation on curcumin (e.g., definition, chemical structure, bioavailability and so on) is needed in introduction or/and discussion.

-Introduction and discussion have been poorly written, which should be improved for the research quality.

-In a method or a result, more detailed explanations on NRS and VASe are required for readers to understand and clarify results of a scale for rating pain.

-In Table 4 and 5, it would be better to simply and clearly describe main outcomes using particular symbols (e.g., ↓, ↔, ↑) rather than a long statement on main outcomes.

-In Table 6, I would suggest to symbols rather than writing low, high and so on.

Author Response

  1. The introduction has been revised as recommended. The reference list has also been adjusted accordingly.
  2. The introduction has been revised as recommended.
  3. We would like to mention that the focus of the present systematic review was to assess randomized controlled trials that evaluated the effect of curcumin in reducing self-rated pain-levels in the orofacial region. The primary focus was the overall reduction in pain and the comparison of pain measurement scales was beyond the scope of the present systematic review
  4. Tables 4 and 5 have been revised as recommended
  5. In table 6, symbols have been used as recommended

Reviewer 2 Report

The authors presents a thorough and well-conducted review relating to the impact of curcumin supplementation on pain levels in the orofacial region. A total of 19 randomized controlled trials were systematically reviewed for the management of pain using curcumin, with the risk of bias assessed using the Cochrane risk of bias tool. The findings were important and significant as they indicated that the vast majority of RCTs suggest curcumin does alleviate pain in the orofacial region.

As such, I believe this manuscript is of high interest to the readership of IJERPH. The manuscript is well presented through data presentation in 6 key tables and cites over 50 relevant works. I believe this manuscript can be accepted in current form.

Author Response

Thank you for your time and valuable feedback. We are pleased to learn that you consider our manuscript acceptable;e for publication in its current form.

Reviewer 3 Report

The aim was to systematically review randomized-controlled trials (RCTs) that assessed the effectiveness of curcumin in reducing self-rated pain-levels in the orofacial region (OFR). The pattern of the present systematic review was customized to primarily summarize the pertinent information. Nineteen RCTs were included. Results from 79% of the studies reported that curcumin exhibits analgesic properties and is effective in reducing self-rated pain associated within the OFR. Three studies had low risk of bias, while nine and seven studies had moderate and high risk of bias, respectively. Curcumin can be used as an alternative to conventional therapies in alleviating pain in the OFR. However, due to the limitations and risk of bias in the aforementioned studies, more high-quality RCTs are needed.

The article is clinically important. I would like to give some comments to improve its quality:

First, it is better to state which databanks have been used for literature search in the abstract.

Second, the benefits of curcumin on other aspect of health related issues should be addressed in the introduction. Please consider to reference this recently published research: https://pubmed.ncbi.nlm.nih.gov/34959636/

Third, please consider to add a PICO question in the method portion in accordance to the PRISMA guideline.

Fourth, in line 114, I would suggest the authors to detail what outcome they would like to examine.

Author Response

  1. Databanks have been added to the revised abstract.
  2. Benefits of curcumin have been addressed in the revised introduction.
  3. A PICO question has been added to the revised method portion
  4. In the revised manuscript, it has been clarified that the main study outcome was reduction in self-rated pain levels.

Round 2

Reviewer 1 Report

Manuscript has been improved for the publication.